# Towards Efficient Battery Electric Bus Operations: A Novel Energy Forecasting Framework

Samuel Würtz [1,]*[ID], Klaus Bogenberger [2][ID], Ulrich Göhner [1][ID] and Andreas Rupp [1]

1 Institute for Innovative Automotive Drives, University of Applied Sciences Kempten,
87435 Kempten, Germany
2 Chair of Traffic Engineering and Control, Technical University of Munich, 80333 Munich, Germany
* Correspondence: samuel.wuertz@hs-kempten.de

**Abstract:** As the adoption of battery electric buses (BEBs) in public transportation systems grows, the need for precise energy consumption forecasting becomes increasingly important. Accurate predictions are essential for optimizing routes, charging schedules, and ensuring adequate operational range. This paper introduces an innovative forecasting methodology that combines a propulsion and auxiliary energy model with a novel concept, the environment generator. This approach addresses the primary challenge in electric bus energy forecasting: estimating future environmental conditions, such as weather, passenger load, and traffic patterns, which significantly impact energy demand. The environment generator plays a crucial role by providing the energy models with realistic input data. This study validates various models with different levels of model complexity against real-world operational data from a case study of over one year with 16 electric buses in Göttingen, Germany. Our analysis thoroughly examines influencing factors on energy consumption, like altitude, temperature, passenger load, and driving patterns. In order to comprehensively understand energy demands under varying operational conditions, the methodology integrates data-driven models and physical simulations into a modular and highly accurate energy predictor. The results demonstrate the effectiveness of our approach in providing more accurate energy consumption forecasts, which is essential for efficient electric bus fleet management. This research contributes to the growing body of knowledge in electric vehicle energy prediction and offers practical insights for transit authorities and operators in optimizing electric bus operations.

**Keywords:** electric buses; energy consumption forecasting; public transportation electrification; auxiliary power models; propulsion power analysis; data analysis

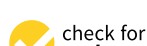



## 1. Introduction

In recent years, the public transportation sector has witnessed a significant shift toward sustainability, with the adoption of battery electric buses (BEBs) emerging as a leading strategy [1]. This transition, while promising in reducing greenhouse gas emissions and improving urban air quality, presents the challenge of effectively managing the energy resources of BEBs to optimize their operation [2]. The energy consumption of electric buses is subject to variability due to factors such as route characteristics, passenger load, and environmental conditions [3]. Therefore, a precise forecast of energy requirements is essential for efficient route planning and optimization of charging locations and battery size [4] as well as vehicle scheduling [5,6]. With means of those energy forecasting models, future scenarios for electric buses can also be investigated; for example, inductive charging at intersections [7], at terminal stops [8], or along the route [9]. With advances in technology, electric buses have demonstrated remarkable efficiency and adaptability in various terrains and climates. For example, an electric truck showed promising results in a field test through the South Tyrolean Dolomites, showcasing the potential of electric buses in challenging topographies [10]. Moreover, studies have shown that electric buses

are economically competitive and can have a lower total cost of ownership compared to diesel buses, particularly when considering societal costs and environmental benefits [3,11]. There are many case studies all around the world. While the number of BEB projects in China is very high, the number of projects in Europe and the United States is growing as well [12–21]. However, reliable integration of BEBs into urban transportation systems necessitates innovative methodologies for accurate energy consumption forecasting. While the average energy consumption of electric buses is around 1.3 kWh/km, this can vary significantly based on operational conditions [22]. Previous studies have made considerable progress in this domain, yet there remains a gap in developing forecasting models that can adapt to a wide range of real-world operational conditions [23]. This study aims to bridge this gap by introducing a novel forecasting methodology that combines a detailed propulsion and auxiliary power model with an environment generator (EG). This approach is unique in its ability to simulate realistic operational scenarios, thus enhancing the accuracy of energy consumption predictions. Our methodology addresses the main challenges in electric bus energy forecasting, which includes estimating future operational conditions such as weather conditions, varying passenger loads, and dynamic traffic patterns. By incorporating data-driven models and physical simulations, we offer a comprehensive tool to understand and manage the energy demands of electric buses under diverse operational conditions.

With this methodology, we aim to answer the following research questions:

- Can data-driven approaches improve energy forecasting for battery electric buses over constant value assumption for practical applications?
- How big is the error margin for data-driven models versus constant values?
- How can bus operators benefit from more precise energy forecasting?

The rest of the paper is organized as follows: Background and Motivation: This section delves into the current landscape of BEB energy consumption research. It outlines existing methodologies, highlights their limitations, and establishes the need for our innovative forecasting methodology. Data Analysis: Here, we present an in-depth analysis of the operational data collected from BEBs. This includes the methods of data collection, preprocessing, and a discussion on the key influencing factors such as elevation gain, temperature, passenger load, and traffic patterns. Methodology: This core section describes our forecasting methodology. It comprehensively details the propulsion and auxiliary energy models and introduces the environment generator (EG) concept, which provides the energy models with estimated input values, such as outside temperature, passenger volume, and traffic conditions. This section explains how these three components synergistically predict energy consumption. Results and Discussion: We present the validation results of our models, comparing their performance and discussing their practical implications in the context of electric bus fleet management and optimization. We also show the potential of the presented approach with two concrete scenarios for bus operations. Conclusions: The final section synthesizes our findings, reflecting on our contributions to the field and suggesting directions for future research. Each section of the paper builds upon the previous, concluding with a comprehensive understanding of the challenges and solutions in forecasting energy consumption for BEBs.

## 2. Background and Motivation

The transition to sustainable public transportation systems, particularly via the adoption of battery electric buses (BEBs), has gained significant interest in recent years. This shift presents unique challenges, notably in the domain of energy consumption forecasting, which is crucial for effective fleet management. The literature relevant to this study can be broadly categorized into three main areas: data analysis, simulation approaches, and prediction models. Each of these areas offers insights into the methodologies and tools used for optimizing electric bus operations, yet they also highlight the need for more comprehensive and adaptable solutions. To understand these challenges, we first examine empirical studies on BEB operations. Focusing on real-world data, these studies offer

insights into energy consumption patterns under various operational conditions. As we explore these insights, we highlight the need for predictive models that can adapt to diverse real-world scenarios.

### 2.1. Data Analysis in BEB Operations

Empirical studies on Battery Electric Buses (BEBs) focus on real-world data to analyze energy consumption patterns. These studies typically examine factors such as passenger load, route characteristics, and environmental conditions. The insights from these studies are crucial for developing baseline models and understanding the practical challenges city bus operators face. In this section, we provide a summary of those recent studies.

Research across different geographical locations employs varied methodologies. Experimental studies were conducted in Belgrade, Serbia [24]; Saskatoon, Canada [25]; and Sao Paulo, Brazil [26], with emphasis on factors like heating, cooling, and driving style. Mišanović et al. [24] found that heating and cooling are key factors for the energy consumption of BEBs in Belgrade. They also reported that driving style could impact consumption by up to 35%. Clarke et al. [25] found similar results in Saskatoon (Canada), emphasizing the dominant effect of heating, ventilation, and air conditioning (HVAC) usage on the range of the vehicles. They also noted that the bus operated reliably in a temperature range of −39 °C to +39 °C. Eufrásio et al. [26] conducted a study in São Paulo and reported an average consumption of 1.19–1.27 kWh/km, of which they found higher energy consumption on hotter days. Different studies provide statistical analyses on energy consumption in relation to temperature [27,28]. Culik et al. [27] performed a statistical analysis on 14,888 trips and found that lower temperatures increased energy consumption. Hao et al. [28] analyzed 197 cars over 12 months and found that below 10 °C, energy consumption increased by 2.4 kWh/100 km for every 5 °C drop in temperature.

Martin et al. [29] conducted a comprehensive study on the impact of road gradient and passenger loading, concluding that the road grade only has a major effect on energy consumption when the passenger load is changing as well. Further, Fernandes et al. [30] reported on propulsion and regeneration efficiencies via an experimental study and found propulsion efficiency to be around 93% and regeneration efficiency around 78%. Vehviläinen et al. [31] observed higher energy use, with winter consumption being 40–45% higher than in summer in Finland, and Wang et al. [32] analyzed sensor data from 99 electric buses across seven cities in China and found that higher average speed increased efficiency. De Wilde [33] and Zhou et al. [34] focused on the impact of air conditioning and passenger load in Brussels and Macao, respectively. De Wilde [33], within their study in Brussels, found that higher temperatures, especially above 22 °C, increased energy consumption due to air conditioning.

Moreover, Zhou et al. [34] utilized onboard diagnostics and a local power company's monitoring system to assess the energy consumption of BEBs in Macao, China. The study found that the average consumption for a 12 m bus is around 1.3–1.7 kWh/km. They also identified HVAC as the greatest impact factor with a 21–27% increase in harsh conditions, while passenger load has a significantly lower effect. They found that from a life-cycle perspective, fossil fuels become reduced by 32–46% by utilizing BEBs instead of diesel buses. These findings underscore the effectiveness of BEBs in dense urban environments and support the argument that BEBs are a viable replacement for diesel buses in such settings. In addition to these studies, He et al. [35] collected data from conventional diesel buses in Knoxville, Tennessee, to establish a framework for evaluating the feasibility of bus electrification, taking into account real-world routes, vehicle performance, and energy consumption patterns. This framework, highlighting an average battery consumption of 1.35 kWh per kilometer, demonstrates the potential for flexible adaptation to various operational scenarios, including differing charging schedules and routes.

These studies help in understanding the special cases in real-world conditions. The analyzed areas either have hot summers (e.g., Brazil) or cold winters (e.g., Norway), which influences whether the heating or cooling has a bigger impact on the overall energy

demand. However, most of the presented studies do not make the leap to build prediction models from the data and do not try to predict environment variables for future bus trips. Building on the empirical analysis, our focus shifts now to the development of predictive models. These models range from simple average consumption estimations to sophisticated machine-learning techniques. Here, we discuss the evolution of these models, their capabilities, and limitations, underscoring the gap in balancing model complexity with practical application.

### 2.2. Predictive Models for BEB Energy Consumption

The modeling of BEB energy consumption is an active field of research. This area has seen the development of various predictive models, ranging from simplistic approaches that use average energy consumption values to sophisticated machine learning algorithms. Some models try to physically model the aspects influencing energy consumption [36,37], while others use deep learning and other data-driven approaches to achieve precise energy forecasting [38,39]. However, there is a gap between the simple models that are used in real-world scenarios and the sophisticated models that cannot be readily applied to the planning process of bus operators.

The broader spectrum of EV energy modeling, as reviewed by Chen et al. [40], is vital for the adoption of battery electric transport systems. This work outlines EV energy consumption modeling trends, particularly the shift towards macroscopic and data-driven models utilizing machine learning on extensive real-world data. It also underscores the necessity for versatile models that extend beyond personal vehicles to include electric buses. Lim et al. [41] provide an in-depth review of different predictive models. Zhou et al. [38] analyzed operational sensor data from buses in Changsha, China, and found that Long Short-Term Memory (LSTM) models outperformed Artificial Neural Network (ANN) models in most cases. Hjelkrem et al. [42] analyzed buses in Norway and China, focusing on propulsion and auxiliary models. They employed a gray box approach but did not validate the predictions against measured values.

Ji et al. [39] used a regression model on real tracking data from Jilin, China; using a temperature Range from $-27\,°C$ to $35\,°C$, they achieved a Mean Absolute Percentage Error (MAPE) of 12.1%. Li et al. [43] used a physical model combined with a CatBoost decision tree model and found that their fusion model had an average error of 6.1%. Basma et al. [44] developed a comprehensive energy model and validated their results in simulations. The applicability to planning for unknown environmental variables is unclear. AlOgaili et al. [45] developed a model to estimate energy consumption, considering integrated elevation data, and validated the model with data from buses in Malaysia. They also do not consider how to use the models for uncertain future scenarios. Pamua et al. developed models in [46,47] using a deep learning network model and found high accuracy on historical data.

A more in-depth review of the different prediction models is not in the scope of this work, and we refer the interested reader to [40,41,48]. The authors of [48] also conducted a review on the methods for estimating the energy consumption of BEBs and found that one of the major challenges in energy prediction is the replication of real-world data.

To conclude, no matter how accurate the prediction model is, its accuracy is limited by the quality of the input data, which can be addressed in different simulation approaches. Lastly, we delve into simulation-based studies. These studies provide valuable predictive insights via virtual modeling of BEB operations under various scenarios.

### 2.3. Simulation Approaches for BEB Energy Prediction

Simulation-based studies contribute significantly to the applicability of the energy models by creating virtual models of BEB operations. These models can simulate various scenarios to estimate energy requirements, including route topographies, traffic conditions, and passenger loads. Although these simulations offer valuable predictive insights, they often require extensive data input and may not fully capture the unpredictability of real-

world conditions. Many studies focus on simulating the vehicle trajectories via traffic simulation [49], driving cycles [50], or assumptions about speed profiles and vehicle mass [51] to estimate the propulsion energy demand. While this is especially important when considering different recuperation efficiencies, the energy demand for HVAC should not be neglected.

Since it is very difficult to predict all input parameters for the HVAC systems, some works do not rely on statistical evaluation of measured data but use simulations to generate input for the models and outside temperature. Wu et al. [52] developed a vehicle model fed with constant speed. They analyzed the effect of different vehicle weights and outside temperatures. However, it is unclear how the data was validated. In the article [53], the authors developed a simulation method to generate bus trips from existing data but used constant values for auxiliary power demand, which makes up a significant portion of the overall demand. Lajunen et al. [54] used simulations to predict the energy demand of BEBs, focusing on different climate conditions and driver behavior. Budiono et al. [55] conducted a study on urban electric buses, finding that a 100 kW motor and 200 kWh battery are sufficient for city use, carrying 85 passengers over 200 km. Based on simulations and GPS data, their research suggests that electric buses are a cost-effective urban transport solution, especially with government incentives, and have a service life exceeding ten years.

Kivekäs et al. [56] introduced a novel driving cycle synthetization method to generate diverse cycles and passenger numbers for bus routes based on a few measured cycles. Applied to a suburban route in Espoo, Finland, the method's validity was confirmed by comparing the statistical properties of synthesized and measured cycles. Utilizing a validated electric bus simulation model, the study analyzed energy consumption variations in ten thousand synthetic cycles of a battery electric bus. Findings included a mean consumption of 0.914 kWh/km, a standard deviation of 0.043 kWh/km, and a consumption variation range of 0.331 kWh/km. This methodology offers valuable insights for public transport authorities, route operators, and bus manufacturers in optimizing bus powertrains and schedules.

Phyo et al. [57] focused on the impact of driver behavior on the power consumption of electric buses in Thailand. The study investigated scenarios, including the critical case of crossing the Rama IX bridge, and explored how climbing resistance affects power demand and recuperation. The research found that higher bus speeds increase power demand, but consumption depends on acceleration and deceleration rates. Notably, faster deceleration or downhill travel can lead to energy recuperation. The study also highlighted that power consumption is higher in traffic congestion conditions. Additionally, the impact of varying speed region lengths on power consumption was evaluated. These insights are crucial for understanding the efficiency of electric buses under varying traffic conditions. The Matlab simulations focused on the speed profiles of the vehicles. Blades et al. [58] conducted a simulation study and found that hydrogen fuel cell buses showed higher range and operating time compared to battery electric buses. In Finland, a computational tool was developed and tested using standard test cycles on two 100 km routes in the center of Manhattan. This tool, which supports operational planning and route optimization for BEBs, reported an energy consumption of 170–200 kWh over 10 h of operation, showcasing the utility of simulation-based tools in real traffic conditions [59].

The existing body of literature on the energy requirements of BEBs highlights a research gap in forecasting methodologies that account for diverse and dynamic city bus operations, including the difficulty of predicting weather conditions and HVAC energy consumption while also being easily applicable to planning purposes of the bus operators. While current simulation approaches serve their purpose, they often overlook some of these critical factors on how to obtain the input data for energy models. This research seeks to bridge this gap by developing an integrated approach, leveraging empirical data analysis, simulation techniques, and advanced predictive modeling to offer city bus operators robust and practical predictive capabilities. This will be achieved via the environment generator (EG), which will be described later in more detail.

## 3. Data Analysis

Before the methodology is described, this section gives an in-depth view of the data used for this study and how it was obtained, processed, and analyzed. The foundation of our study is a comprehensive dataset derived from high-resolution operational metrics of 16 Mercedes E-Citaro articulated electric buses operating in Göttingen, Germany over a 1-year period. This dataset includes granular information such as GPS data, acceleration, speed, vehicle load, interior and exterior temperatures, energy demand details, and more, as documented in the OMNIplus On data interface www.omniplus.com/de/on/ (accessed on 11 January 2024).

### 3.1. Data Collection and Preprocessing

Our data collection strategy focused on capturing a complete picture of BEB operations. We extracted approximately 45,000 individual bus trips between terminal stops. Each trip was analyzed to gather metrics necessary for estimating energy demand. This included information on the line and vehicle specifics, start time and date, trip duration, location coordinates, temperature averages, passenger counts, and state of charge (SOC) levels of the vehicle's battery.

Since the observed vehicles primarily served two different lines, the analysis was focused on those. All the trajectories were split at the terminal stops of those lines (see Figure 1). For every single detected trip, the following metrics were collected: the line, vehicle, start time and date, duration, start and end location, mean temperature, mean and maximum passenger count, the passenger-kilometers, mean and max speed, number of stops (every time the vehicle slows down to 0 km/h, not differentiating the causes of stopping), electrical energy, energy for the auxiliaries (mostly HVAC), and start and end SOC (state of charge). The two analyzed lines cover a distance of 11 km and 13 km, respectively, and take around 30 min. Since the detection algorithm is imperfect, we had to filter out some trips with unrealistic values, where the served distances exceeded the actual distances, which was usually caused by changes in the bus schedule when the bus did not reach the geo-fence for the terminal station.

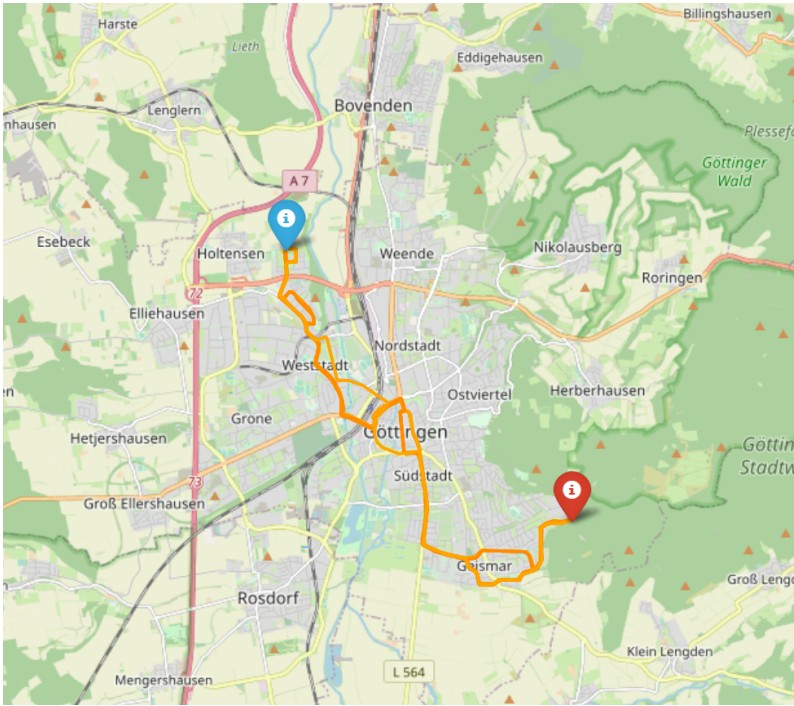

**Figure 1.** Visualization for one bus on one day, serving the same line the whole day.

Figure 2 is an example trajectory between two terminal stops, showing the variability of the altitude profile, typical speed, and passenger load profiles. One of the major impact factors for the energy demand is the altitude change, which can be seen there. The effect of the elevation change can also be seen in Figure 3, which shows a histogram of the energy demand of all recorded bus trips. The different peaks indicate different lines. While the middle peak, around 1.55 kWh/km, represents the bus line with almost no elevation gain, the lower and upper peaks represent the bus line with an almost 150 m elevation difference between the terminal stops. The peak around 0.7 kWh/km represents the values for the "down-hill" direction, and the high peak at around 2.2 kWh/km represents the "up-hill" direction. The dotted red line indicates the energy demand for planning as defined by the OEM.

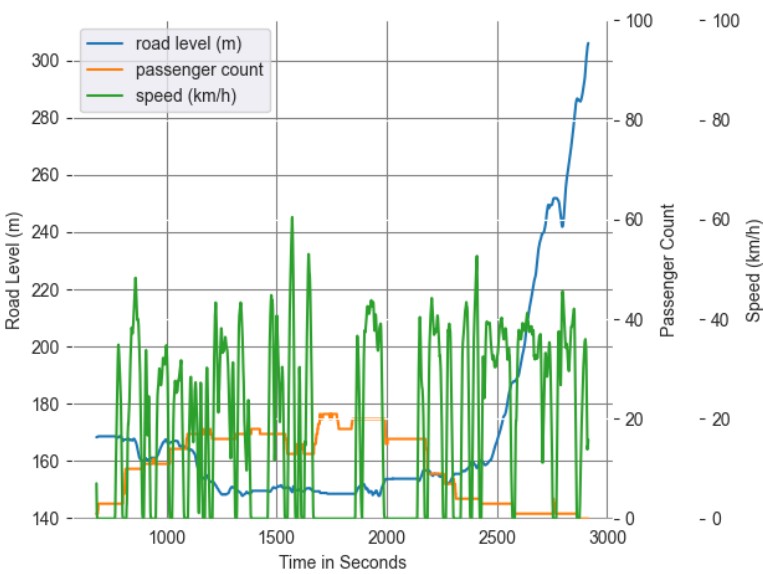

**Figure 2.** Example of the tracking data depicting passenger count, speed, and altitude.

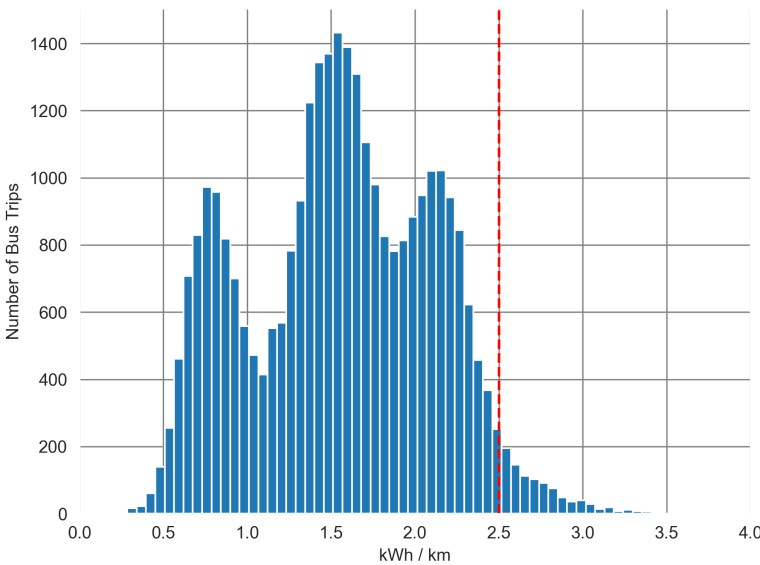

**Figure 3.** Histogram of the energy consumption on every trip recorded with three distinct peaks, indicating different bus lines.

In order to compare the different trips, the average energy consumption was adjusted based on the elevation gain between terminal stops based on the potential energy. The

idea was to be able to analyze the trips as if they all happened on flat terrain. A simple formula was used to calculate the potential energy difference according to elevation changes. $e_{pot} = m \times g \times h$, where $m$ is the weight of the vehicle, $g$ is the gravitational constant of the Earth, and $h$ is the elevation change. Neglecting passenger weight and considering the curb weight of 20,000 kg and the elevation change on one line of 26 m and the other line of 144 m, the correction values are 1.4 kWh and 7.8 kWh for the whole line. This results in an only slightly skewed normal distributed energy demand histogram as seen in Figure 4. The dotted red line in this figure also indicates the energy demand for planning as defined by the OEM.

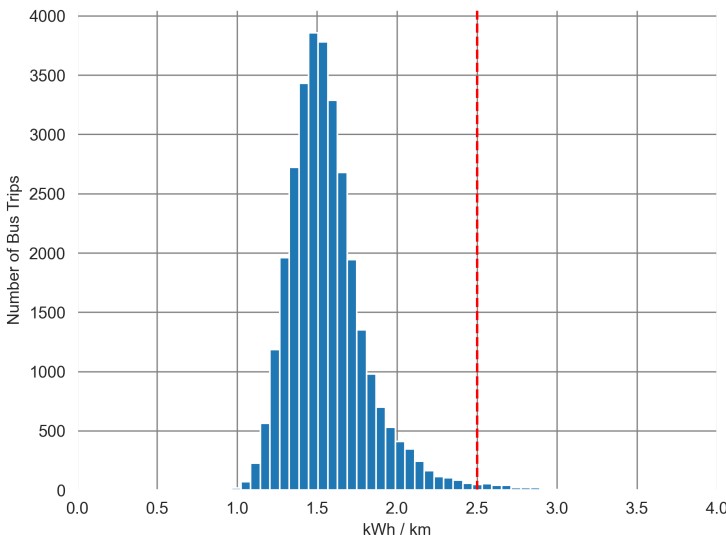

**Figure 4.** Histogram of the energy consumption on the bus lines after applying the elevation gain energy correction.

After applying the filtering of unrealistic trips, calculating and eliminating the influence of the elevation gain-induced potential energy, and adding information about the time of the day and year, the data was ready for further analysis. In order to obtain a better understanding of the dataset, basic numbers were extracted and visual tools like histogram plots were used to identify distinct patterns in the energy consumption of other analyzed BEBs. Additionally, we employed statistical methods to evaluate the influence of temperature on auxiliary power usage and passenger load on both tractive and auxiliary energy demands. These analyses were critical in understanding the interplay between operational variables and energy consumption.

*3.2. Basic Numbers*

This section provides an overview of the data used for this study. The vehicles used are 16 Mercedes E-Citaro articulated electric buses.

Vehicle Specifications: The analyzed E-Citaro bus is equipped with dual drive axles, offering a total rated power output of 504 kW. The HVAC system includes a 22 kW heater and an 8 kW cooling system in the front, alongside a 36 kW water-circuit heater and dual heat pumps providing 17 kW heating and 23 kW cooling in the passenger area. Additionally, the bus has 14 kW floor heating and a supplementary 23 kW diesel heater, which is predominantly used when temperatures are below 8 °C. The maximum HVAC power usage is 54 kW for cooling and 129 kW for heating.

The data was collected from two bus lines where the vehicles approximately serve the lines 10 times in each direction. Both routes go straight through the city center of Göttingen, Germany. They vary significantly in their topology.

Operational Data Overview:

- Total Trips Observed: The analysis is based on 46,675 bus trips from 16 vehicles over 13 months. From November 2022 to November 2023.
- Total Electrical Energy: A cumulative consumption of 619 MWh was recorded.
- Auxiliary Energy: Auxiliary systems accounted for 176 MWh, 28.5% of the total consumption. This appears to be quite high considering the additional diesel heating for cold conditions.
- Passenger Kilometers: The buses covered 6.82 million passenger kilometers (pkm).
- Electrical Energy per Passenger Kilometer (kWh/pkm): The average consumption was 0.09 kWh per passenger kilometer.
- Temperature Range: Operational temperatures varied from −12 °C to 33 °C.
- Passenger Volume: The average number of passengers was around 19, occasionally exceeding the maximum capacity of 145 passengers during peak hours.

Although the bus operators know the operations very well, they are very cautious in the utilization of the battery capacity. Notably, the State of Charge (SOC) generally remained above 70%, indicating a conservative battery usage pattern in these vehicles (see Figure 5). This indicates that advanced charging strategies might have a big potential on the economic performance of the BEBs.

These basic numbers lay the groundwork for our comprehensive analysis, providing insights into the energy dynamics of battery electric buses.

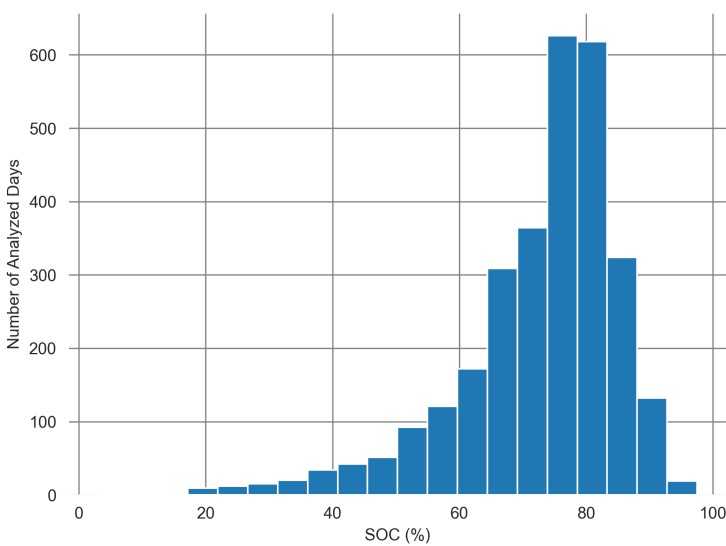

**Figure 5.** Histogram of the lowest SOC of the day.

### 3.3. Influencing Factors

This section describes the major influencing factors on the energy demand. To reduce the noise in the distributions and obtain a clearer picture of the different factors, the energy demand for propulsion and auxiliary components are analyzed separately. We used basic statistical tools, for example, calculating the mean, min, and max values for the different variables, as well as different percentiles of the data to better understand the distribution of values. The focus is on the propulsion energy demand since the elevation, traffic, and driver behavior mostly influence the tractive forces. The passenger load influences the tractive and auxiliary energy demands because of the passengers' additional weight and thermal output. The outside temperature only influences the auxiliaries, mostly the air conditioning/heating of the passenger area and the battery temperature control. In the following sections, the different variables are described in more detail.

### 3.3.1. Elevation Gain

As mentioned in Section 3.1, elevation change greatly impacts the energy demand of a single bus line. In order to analyze the other effects in more detail, the described elevation gain adjustment was employed. This way, statistical analysis over a large number of trips was possible. According to our evaluations, with a sufficiently powerful electric motor and batteries, most potential energy can be recuperated and used to charge the batteries, minus the efficiency losses of the engine. Since most vehicles will travel the same elevation up and down during the day, it should even out. However, there are some aspects to consider; if a bus operator decides to acquire vehicles with smaller engines, there might be significant losses when the machine runs into power limits in generator mode; if the vehicle travels uphill with many passengers and downhill almost empty, the assumption made will not hold, since the potential energy changes with the vehicle mass, as it was shown in [29]. However, due to the high amount of data evaluated in the presented paper, it can be assumed that such extreme fluctuations in the passenger load have a less significant impact as they should approximately equalize over the observation window.

In the case of the analyzed routes in Göttingen, the altitude changes happen either at the beginning or at the end of the bus line, where the passenger load is the lowest (see Figure 1). There is a significant difference in energy consumption of the uphill and downhill routes, where the downhill route has an average consumption of 0.7 kWh/km and the uphill route 2.2 kWh/km (see Figure 3). Since potential energy forms a big part of the overall energy demand, charging strategies should also consider elevation changes. If the charging station is on top of a hill, the vehicle should not be charged fully; otherwise, the potential energy will be lost and converted to heat via the mechanical brakes.

### 3.3.2. Temperature

We identified the second largest influencing factor using the tracking data as the outside temperature. As shown in Section 3.2, even though an additional diesel heater was used, the auxiliary power still accounts for more than 28% of the total energy demand, which is mostly due to the HVAC system.

In Figure 6, a scatter plot is shown where each individual bus trip is represented by one blue dot. Each blue dot is the average auxiliary energy demand of that trip, and on the *x*-axis is the average outside temperature, which reveals an interesting pattern. A clear dependence of the auxiliary energy demand and the outside temperature is visible, while the lower temperatures are split into two cases: (a) pure electric heating (the upper branch) and (b) heating with the supplementary diesel heater (lower branch). Unfortunately, in this study, we did not have information about the state of the diesel heater, which makes the prediction tasks a bit more challenging.

Two distinct distributions become visible by analyzing the histogram of the auxiliary energy demand in Figure 7 of temperatures below 10 °C. Therefore, we decided to add estimated labels on whether the diesel heating is on or off during a trip. To obtain this information, a Gaussian mixture model with two components was fit onto the data of trips with an average temperature below 10 °C. Trips with higher temperatures were labeled as "off" for the additional heater. This information was used later on for building prediction models. Although there is some uncertainty with the correctness of the labels around 0.8 kWh/km, predictions for those energy demand values could either be from case (a) or (b) (with or without additional diesel heating). Aside from that, it becomes apparent that the outside temperature significantly impacts the overall energy demand.

This shows that the local and seasonal temperatures should be considered strongly when planning bus operations. Additional diesel heaters can be considered for the ramp-up phase, especially for very cold regions. Still, in pursuit of zero-emission public transport in the long term, the vehicles should be fit to power the HVAC as fully electric.

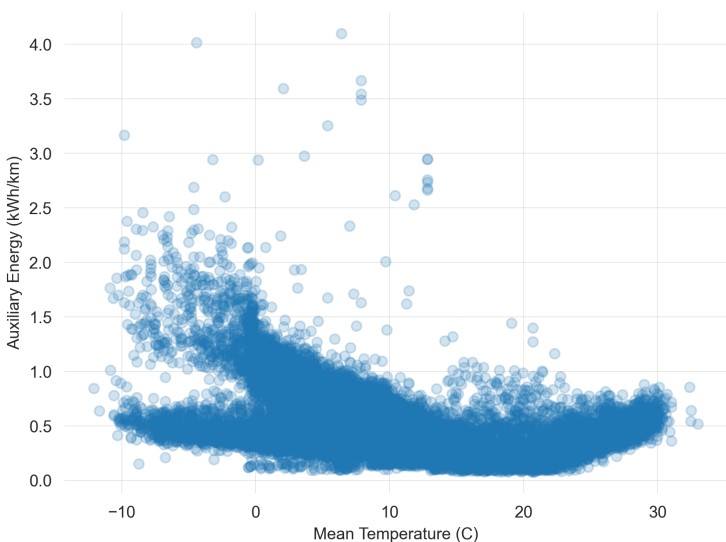

**Figure 6.** Blue dots' average represent the auxiliary energy demand for a bus trip at a given mean temperature.

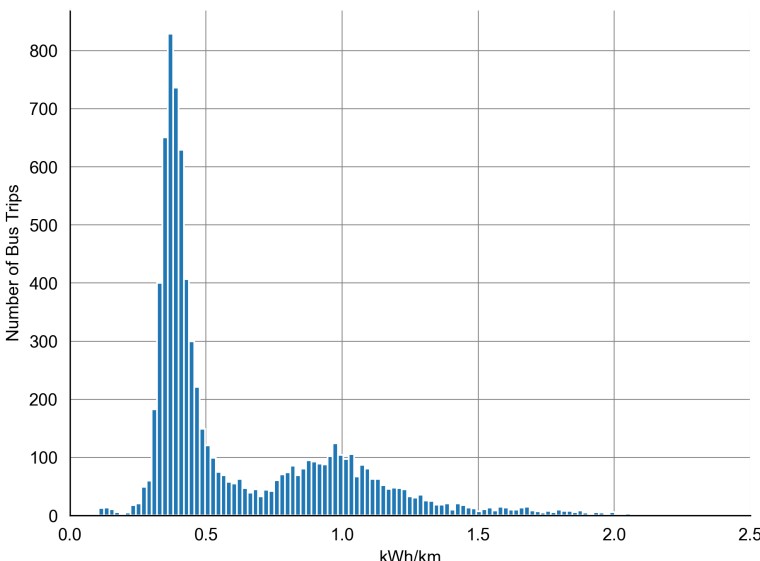

**Figure 7.** Histogram of the auxiliary energy demand on the different bus trips. For all trips with temperatures below 10 °C.

### 3.3.3. Passengers

The passenger load is also a significant factor in the planning of BEB operations. It will mainly influence the tractive forces in areas with big elevation changes, but also impacts the HVAC system.

In Figure 8, a clear trend is visible where more passengers mean higher average energy consumption, with an increase of more than 20% from an almost empty bus (average passenger count below 10) to a rather full bus (average passenger count above 75). It is important to note that other factors still heavily influence the expected energy demand, which is shown in the box plot by the large value range (1.5 × Inter-Quartile-Range (IQR)) indicated by the lines above and below the colored bars.

In Figure 9, the impact of passenger numbers on the auxiliary energy demand is shown, which is much less than the effect on the propulsion energy demand. Ignoring the outlier with only two observations for more than 75 passengers, an increase of 8%

(0.36 kWh/km to 0.39 kWh/km) can be observed from an empty to a full vehicle. Here, the dominance of other factors over the passenger count is even stronger since the value range is almost the same for every passenger number category.

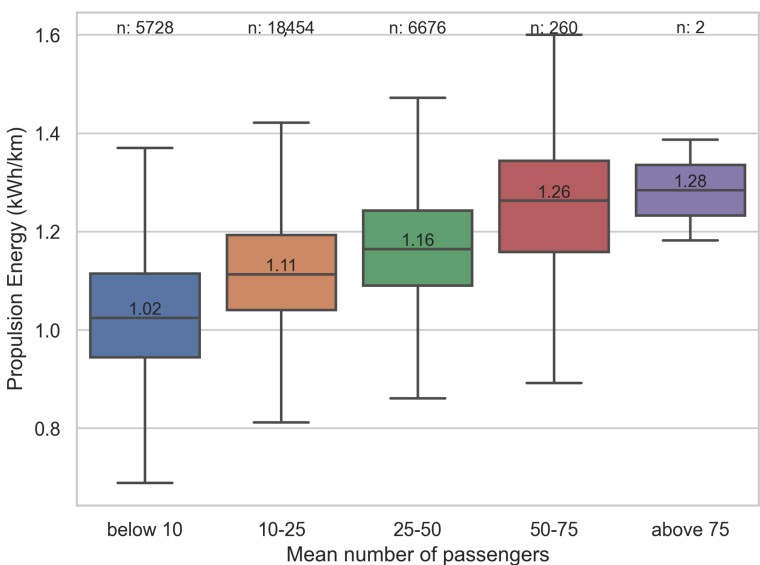

**Figure 8.** Box plot for the average propulsion energy demand for different passenger counts. Median values are annotated in the plot. The number (n) of samples (bus lines) constituting each bar is annotated on top.

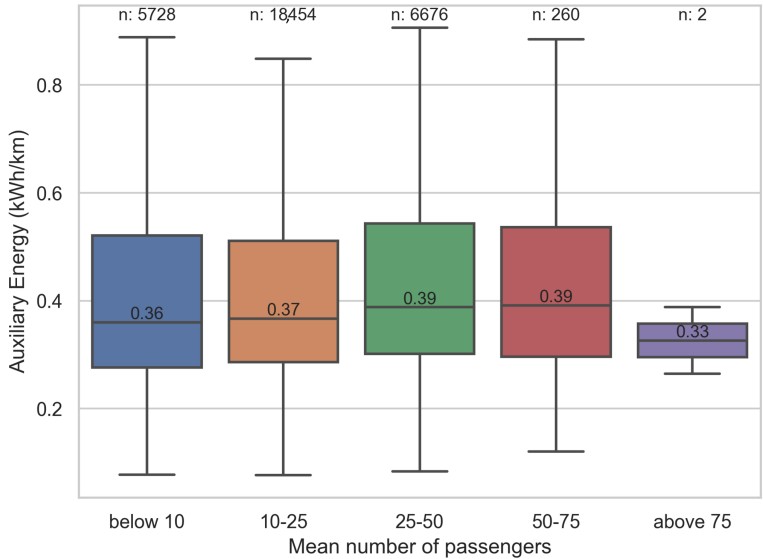

**Figure 9.** Box plot for the average auxiliary energy demand for different passenger counts. Median values are annotated in the plot. The number (n) of samples (bus lines) constituting each bar is annotated on top.

Due to increasing data collection in public transport, the knowledge about the number of passengers on a line at different times of the day is getting better and better. Therefore, it is useful to incorporate such information into prediction models.

### 3.3.4. Traffic and Driver

Another influencing factor is the speed profile of the vehicle, which is influenced by the driver as well as the surrounding traffic. With a data-driven approach utilizing tracking

data, separating the influence of drivers and traffic is very difficult. Different analyses were performed to analyze the effects.

A clear trend is visible when visualizing the propulsion energy demand over the number of stops during the serve of one line in Figure 10. Since it is not differentiated between regular bus stops and traffic-caused stops, the conclusion can only be of a broader nature. However, we could see from our data that more constant driving results in lower energy demand. From the analyzed data, we obtain more than a 40% increase in the average energy consumption for the propulsion energy from 10 to 50 stops for one line serving.

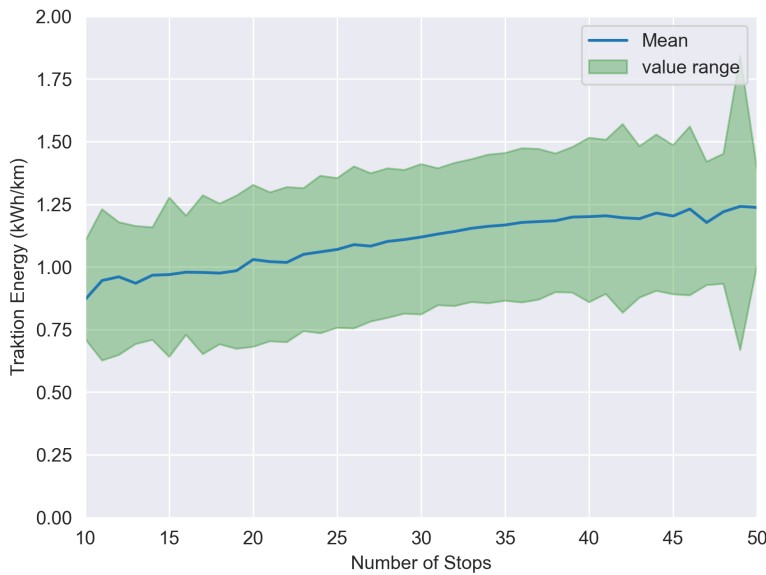

**Figure 10.** The blue line indicates the mean value for the propulsion energy demand, while the green areas indicate the value range.

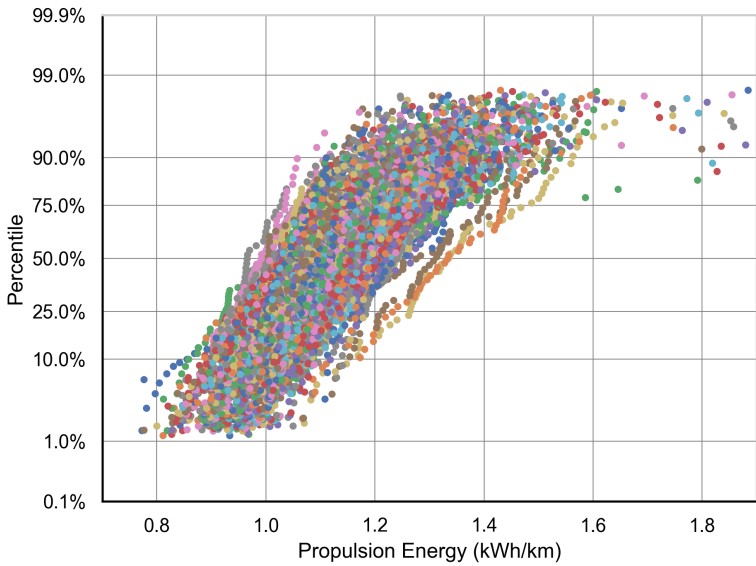

**Figure 11.** Percentile plot of the propulsion energy demand for each driver.

In order to analyze the different drivers, a different visualization is required. Since the number of samples per driver is much lower, a percentile plot was chosen to compare different data samples with different sample sizes and their statistical properties (see Figure 11. On the *y*-axis is the percentile, while on the *x*-axis is the value of interest, the energy consumption. In the middle of the *y*-axis is the median value for the data sample, the

50th percentile. By looking at the x values for the 10th and 90th percentiles, the distribution of the data can be visually understood.

Every driver is plotted with a different color. From the plot, it can be concluded that the variance of the energy demand of the different drivers is quite similar since the slope of each plotted dataset is similar. But, it still seems like drivers with a lower median energy consumption also have a lower variance, shown by the flatter slopes. An increase of around 30% from the most economical driver to the least economical driver can be observed by analyzing the median values. This can inform bus operators about the potential for savings when giving the drivers training in economical driving.

This analysis showcases the importance of understanding the speed profile of the city buses since this significantly impacts the energy demand. Traffic simulation tools can be a good method to obtain realistic results. In future studies, the effect of different drivers could be investigated more deeply, and a model for an optimal eco-driver could possibly be developed.

This comprehensive data analysis sets the stage for our subsequent modeling efforts, where these insights are integrated into predictive models for energy consumption forecasting in BEB operations.

## 4. Methodology

After introducing the background and the dataset, we now dive into the energy prediction framework. The proposed methodology integrates three components into a robust energy predictor for BEBs: a propulsion energy model, an auxiliary energy model, and an environment generator. This section describes this framework in more detail with all its properties and interactions.

### 4.1. Forecasting Framework

The overarching concept of this framework is to validate energy models against measured data and then utilize the same framework for predicting energy demands for future bus trips.

It can be applied to bus trips with measured data from BEBs to evaluate the performance of different energy model instances. Operators can later use the framework for energy predictions for specific bus lines, dates, and times, employing the most effective model found via prior investigations.

In Figure 12, all the different components of the framework are illustrated. The layout of the graphic is roughly based on the notations of a class diagram from the field of software development. This means that the filled arrows indicate the specialization of different classes. In this case, the different energy models are concrete instances of either the auxiliary or propulsion power models. The "normal" arrow indicates the direction of control, where the operator uses the predictor, which in turn uses the EG and the two models.

The content of Figure 12 can be divided into five logical components:

- blue: The actor side, which includes the person using the framework; the front end of the framework, called the predictor; and the results report;
- gray: The input for the whole framework, which consists of the aggregated line data from the buses; the raw tracking data; and the historical weather data;
- purple: The propulsion energy model, which is one of the available instances (Physical Model, Daytime Altitude Model, Constant Altitude Model, or Constant Model);
- yellow: The auxiliary energy model, which is one of the available instances (MLP Model, Temperature Model, Monthly Constant Model, or Constant Model);
- green: The environment generator.

When employing the framework for a specific use case, the best combination of energy models is used depending on the available data. In the simplest case, the constant version is configured for both the auxiliary and propulsion models. If more precise information is needed and the data and know-how are available, a combination of more complex variations in the energy models could be employed. The bus operator only needs to supply

the specific bus line, the date, and the time to obtain the energy demand prediction. The predictor first needs to query the EG to obtain the required environment variables, which in turn queries the required input from the weather API and the bus line data. With the results of the EG data, the predictor proceeds to request the energy demand predictions for auxiliary and propulsion energy from the configured models. Finally, the predictor combines the results and generates the results report for the bus operator.

The historical weather data and bus line data are required for both the model development as well as the energy prediction, while the tracking data is only required for validating the models' accuracy during model development.

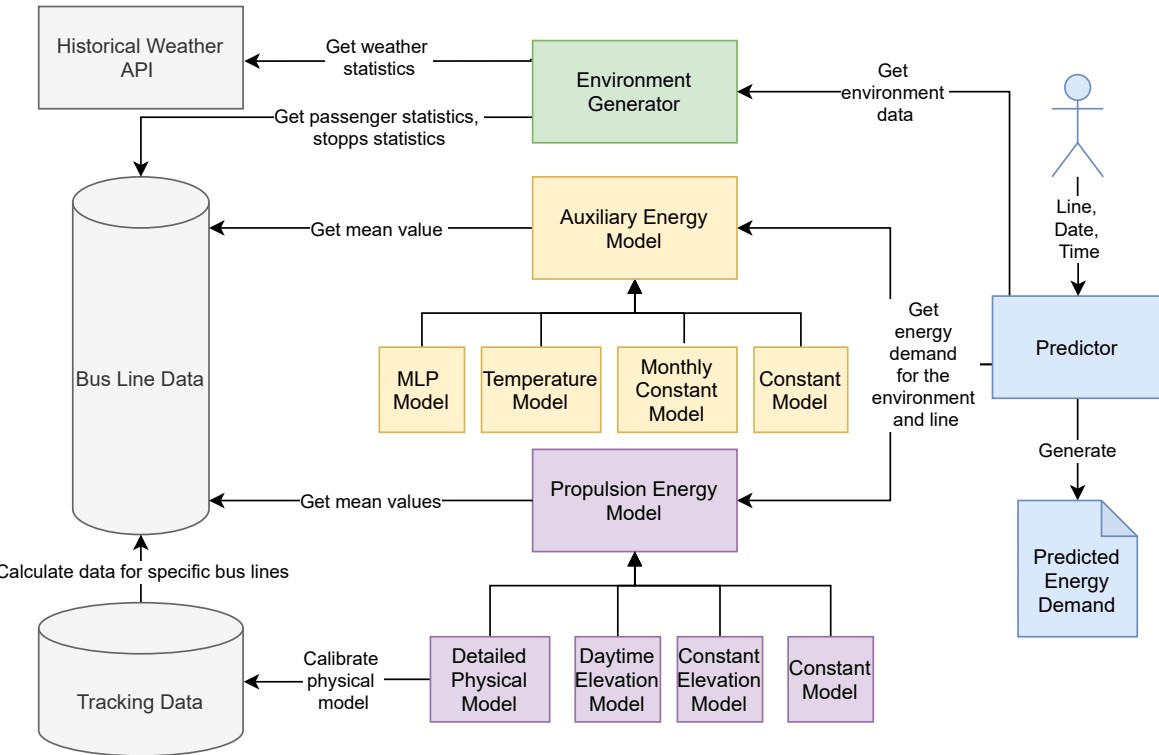

**Figure 12.** Overview of the energy forecasting framework. The gray parts are input data to the framework, the blue parts represent the operator side, while the EG (green) represent the Auxiliary Energy Model (yellow) and Propulsion Energy Model (purple) form the core of the framework.

The following subsections describe the different models, their strengths, and weaknesses in more detail, as well as the functionality and scope of the EG.

### 4.2. Propulsion Model

Predicting the propulsion energy of a battery electric bus involves considerations of known distances and topology, along with variables like passenger volume, traffic, and street conditions.

### 4.2.1. Constant Model

The constant model, the simplest form of energy prediction, requires only an average value for energy consumption per kilometer. This model's advantage lies in its ease of use and broad applicability. However, it does not account for local factors such as topology, potentially leading to significant uncertainties and necessitating large safety margins. Despite its weaknesses, it is still the most widely used too for BEB planning.

### 4.2.2. Constant Model with Elevation Gain

As discussed earlier, elevation gain, depending on the city, can be the most important factor for the propulsion energy demand. It therefore seems obvious to implement a simple

model to incorporate this information. In this work, an algorithm was developed, partly consisting of the previous constant model and, therefore, needs the distance as input. In addition, the altitude of the start and end of a route are required to calculate the potential energy. The only other information required is the mass of the vehicle. For this approach, though, if no additional information about passenger volume is available, it is enough to use the vehicle mass plus an average number of 18 passengers. The average passenger weight of 65 kg was taken from the MAN report [60]. Then, with the formula of potential energy $E_{pot} = m \times g \times h$, the additional energy demand for that trip is calculated, which can be either positive or negative based on the topology, where $m$ is the empty weight of the vehicle + passenger weight, $g$ is the gravitational constant, and $h$ is the elevation gain between the start and the end of the bus trip under analysis. This approach greatly improves accuracy over the constant model for the analyzed routes in Göttingen, Germany.

### 4.2.3. Daytime Model with Elevation Gain

To improve the accuracy further from the last model, there are many factors to consider besides the distance and topology. The more obvious one is the speed profile, which is determined by the driver's behavior and the surrounding traffic. Other factors are the road surface, tire pressure, and even wind. But, the latter ones are exceptionally hard to incorporate, and their effect is presumably lower than the others. The driver behavior is also a variable that should be tackled with driver eco-coaching, and the traffic influence is specific to certain cities and times of the day. This is why, for this model, statistic values for the different times of the day were aggregated and are used for predicting the daytime-specific propulsion energy demand. This approach requires the city to have some buses available to collect the required data since the traffic patterns in different cities might differ greatly. Although it can further improve the results, it will only benefit some bus operators who already have some experience with BEBs. For others, it might not be a practical solution.

### 4.2.4. Physical Model

The physical models, as described in [36], can potentially yield the most accurate results. It was developed with the use case in mind where operators have a fleet of combustion engine vehicles and want to estimate the energy demand of the future buses based on the movement patterns of the diesel buses. In that scenario, the movement of all the buses is recorded utilizing GPS trackers. This information, and other available data like topography and passenger volume, can yield precise results. Therefore, this model is an upper-bound reference for the accuracy of the propulsion energy model. The advantages of this model are that it can be easily applied to new observation areas and can be parameterized to any bus type. The disadvantages are the need for tracking data and some domain knowledge to apply it to new vehicles. Using concrete speed profiles for energy estimation is most likely impossible in classical bus operation planning tasks. While it might be possible to generate speed profiles for different routes and times of the day, it is questionable if the effort is worth the gained accuracy.

In upcoming studies, it could be investigated how close the accuracy of the physical model with simulated input data can get to the physical model with real trajectory input data. This might be an option to improve the accuracy over the daytime model with elevation gain.

### 4.3. Auxiliary Model

The auxiliary energy demand is the other major part of energy forecasting for battery electric buses, which consists of all components requiring energy besides the drivetrain. The auxiliary energy demand is dominated by the vehicle's heating, ventilation, and air-conditioning [61]. Since this part depends on variables like outside temperature, precipitation, number of passengers, sun intensity, number of door openings, and many more, a data-driven approach was chosen in this article. We started with a very simple

constant model. Analogous to the propulsion model, different levels of complexity were implemented, while the most sophisticated analysis was a neural network in the form of a multi-layer perception.

### 4.3.1. Constant Model

Analogous to the propulsion energy model, a simple constant model was set up, which provides only a single value for the HVAC energy consumption. Since HVAC energy demand highly varies through the seasons and daytime, the approach is not optimal, but instead requires an often-applied simplified way to incorporate auxiliary energy demand. It provides a very simplistic tool for bus operators to obtain a first feel for the feasibility of different electric buses on their routes.

### 4.3.2. Monthly Model

The monthly model improves on the constant model by providing monthly consumption values. This can significantly improve the results but is not resilient to outside changes, like the climate in a different city or global warming. That said, it is still a very simple tool for bus operations planning while it achieves higher accuracy than the constant model.

### 4.3.3. Temperature-Based Model

The development of the temperature-based model is an attempt to build a location- and climate-agnostic model for the auxiliary power demand. By only incorporating the outside temperature, the operators can use climate tables or weather forecasts to predict the auxiliary energy demand. This model used all the tracking data from over one year of bus operations. The average values per temperature were calculated and extrapolated to temperature ranges beyond the observed values. Since the HVAC in heating mode is a completely different system than in cooling mode, two different functions were fitted to the data: one for the values equal to and above 20 °C for the cooling portion, and one for the values below 20 °C for the heating portion of the HVAC. By utilizing domain knowledge about the HVAC, this separation is possible and allows for simpler functions.

A box plot was created to show the outside temperature's influence on the auxiliary energy demand (see Figure 13). The middle line inside the box shows the median value, and the colored box shows the IQR from the 25th to the 75th percentile of the observed values. The outer marks indicate the value range which is defined by 1.5 times the IQR range. All values beyond this would be considered outliers. This plot shows a strong correlation between temperature and energy demand. For all values below 10 °C, we can see a much higher variance in the data, indicating the usage of the additional diesel heater in most cases. For predicting the energy demand, the model can be fitted to either the values without diesel heating or the values with diesel heating, reflecting different operating modes.

### 4.3.4. Neural Network Model

Since the auxiliary power demand is influenced by more than just the outside temperature, it was an obvious choice to develop a machine learning-based model incorporating all the easily accessible information. In this case, it was the average temperature during the trip, the time of day, the month, and the average number of passengers during the trip. In the beginning, many more input parameters were used for every trip: the maximum number of passengers, number of person kilometers, average speed, maximum speed, number of stops, and maximum SOC. However, after a thorough parameter search, the additional inputs did not improve the accuracy. All the tested parameters from the parametersearch can be found in Table 1, the best parameters are marked in bold font. In accordance with Occam's Razor, which states that theories should be as simple as possible but not simpler [62], it was decided that the minimum number of inputs that still bring the best results should be used.

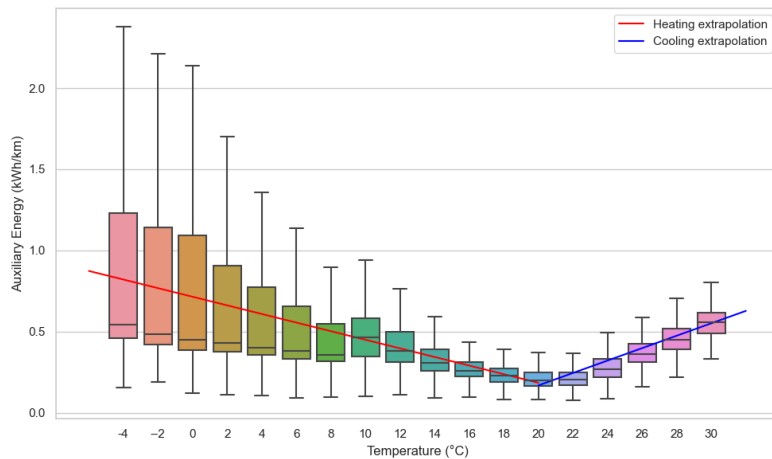

**Figure 13.** Box plot of the auxiliary energy demand for different outside temperatures. The red line shows an estimated extrapolation of the mean values for cold temperatures without the additional diesel heater. The blue line shows the estimated energy demand for cooling the vehicle in temperatures above 20 °C.

**Table 1.** Values for the hyperparameter search of the machine learning model. The best values are marked in bold font.

| Hyperparameter | Values |
|---|---|
| Learning Rate | $10^{-5} \ldots$ **0.0029** $\ldots 10^{-1}$ |
| Layers | [64, 32], [64, 32, 16], **[64, 32, 16, 8]**, [64, 32, 16, 8, 4], [64, 64, 64], [2, 2, 2, 2], [8, 8, 8, 8] |
| Activation Function | relu, tanh, **sigmoid** |
| Optimizer | **adam**, sgd, rmsprop, adadelta, adagrad, adamax |
| Loss Function | mse, **mae** |
| Epochs | **1000** |
| Batch Size | 16, 32, **64**, 128 |

The Neural Network (NN) was trained with an 80-20 test–train split and the Early Stopping callback to find the best weights and prevent overfitting. The input values in the final model were average temperature, average passenger number, maximum passenger, person kilometer, number of stops, and the flag indicating diesel heater usage. The heater flag was obtained using the Gaussian Mixture model as described earlier in Section 3.3.2. The model's accuracy could already be significantly improved by adding the diesel heating flag. We used the accuracy of this model as the best-case scenario for the auxiliary power, since with perfect knowledge of the input parameters, we obtain the best result.

Table 2 provides an overview of the different models' strengths and weaknesses. But, in order to get the best out of those models, good input parameters are required. To obtain those input parameters, we must focus on the framework's third and last part, the EG.

### 4.4. Environment Generator

The environment generator (EG) is a crucial part of this methodology. While the authors of this article developed sophisticated models, the quality of the results mostly depends on the quality of the available input data. This is where input data generation for future bus trips comes into play. This usually gets neglected in existing studies about energy prediction models. The main information required for the prediction are weather conditions, where the temperature is most important, traffic, and the number of passengers. Although more data sources influence energy consumption, in the process of training a neural network for energy prediction, we found that their influence is marginal compared to the uncertainties introduced by estimating the input data. In this first step of the

environment generator, we analyzed the different influencing factors statistically. This could potentially be improved by incorporating more advanced machine learning models.

### 4.4.1. Weather

The weather mainly affects the auxiliary power, not the propulsion energy demand (besides some minor influences via different rolling resistance with rain or snow). The inside temperature of the vehicles is influenced by outside temperature, sun, precipitation, and wind. The weather can also have an impact on traffic since heavy rain and snow can slow down traffic. However, since the different factors have negligible effects on energy consumption compared to the outside temperature, it was chosen to only generate data for the outside temperature. This was performed by means of a historical weather API. In this case, meteostat.com provides an easy-to-use interface to download historical weather data. The dataset includes the date and time, temperature, relative humidity, precipitation, snow, wind direction, wind speed, wind gusts, pressure, and hours of sun. This was used to generate average temperature values for every hour of the year based on the past five years for the given location. For reference on the different temperature ranges, see Figure 14. Values are generated by averaging the values for each day based on the last 5 years.

The EG is asked to provide a temperature value for a given location, date, and time. This can then be used as input for the prediction models.

**Table 2.** Advantages and disadvantages for the different auxiliary and propulsion Models.

| Model | Advantages | Disadvantages |
|---|---|---|
| Auxiliary Models | | |
| Constant Model (CM) | Easy implementation | Low accuracy |
| Monthly Model | Easy implementation, better seasonal accuracy | Region specific |
| Temperature Model | Adaptable to different cities, seasons, and climatic regions | Specific to vehicle types, requires data collection for new vehicles |
| Neural Network | High accuracy | Requires extensive data, vehicle specific |
| Propulsion Models | | |
| Constant Model (CM) | Easy implementation | Low accuracy |
| CM with Altitude (wA) | Accounts for topology | Requires topology data, improved but still low accuracy |
| Daytime Model wA | Considers topology and traffic | Location and vehicle specific |
| Physical Model | High accuracy, adaptable to new vehicles and regions | Data and calibration intensive, requires trajectory data, e.g., from simulations |

### 4.4.2. Traffic

Traffic mostly has an impact on the propulsion energy because dense traffic can force the vehicle to do multiple additional acceleration and deceleration maneuvers. In Figure 10, the energy demand correlates with the number of stops, but predicting the number of stops is quite challenging. A different approach was chosen which tries to incorporate the information that might be used to predict the number of stops: the bus line served and the time of day. In this case, the EG uses the aggregated bus line data to generate adjustment factors for the propulsion energy demand based on the time of day and the bus route that gets served.

This approach has some obvious limitations because the findings do not readily apply to other cities. While the traffic patterns might be similar, they will differ a lot between the bus lines. A different approach would be using traffic simulation tools to simulate the lines at different times of the day. This is also challenging since it is difficult to incorporate realistic traffic information into the simulation. As more and more cities around the world

are adopting digital twins for city planners, it sometimes include traffic simulation tools [63–65]. Traffic simulations could be a viable option for those cities. The validation of the result is an interesting topic for follow-up studies.

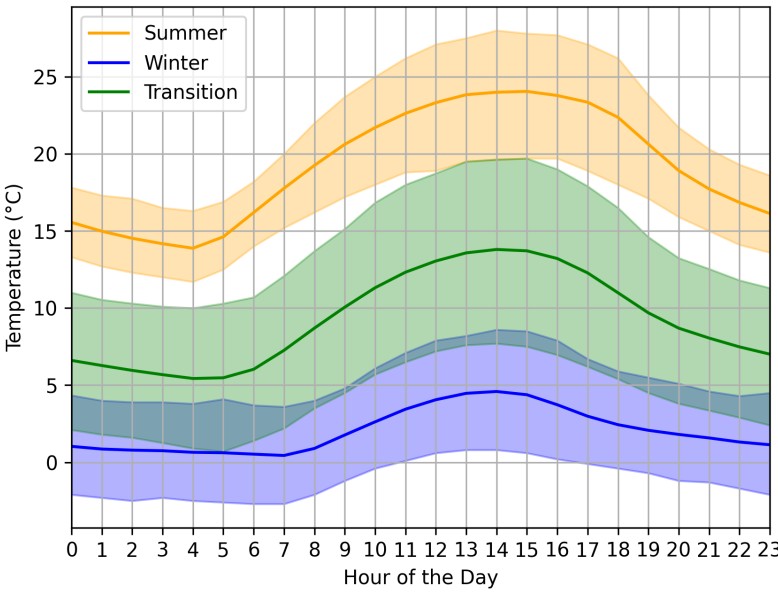

**Figure 14.** Temperature ranges and median values for Göttingen Germany during the day, for different times of the year.

### 4.4.3. Passengers

The data analysis section shows that the passenger volume significantly impacts the energy demand. There is a notable impact on propulsion energy, while the effect on auxiliary power is comparatively minor. To integrate these observations into our energy models, the aggregated bus line data was utilized to obtain statistical representations of passenger volumes at varying times and across different bus routes. Whenever a city has passenger data available for the bus lines, this is a good approach. However, in order to obtain a more generalizable model for different cities, a machine-learning approach could be possible. Variables such as points of interest (POIs), urban size, and time of day could be used to train a machine learning model to generate more accurate passenger volume predictions for the bus lines. This is likely a challenging task but might be worth investigating.

The framework introduced offers a comprehensive structure for predicting the energy demand of BEBs. Although individual components of the framework present opportunities for enhancement, it establishes a modular and robust foundation for systematic improvements. Future models with physical modeling of the auxiliary powers, more detailed environment generators for traffic conditions, or machine learning-based propulsion energy models could be seamlessly integrated into the framework and their accuracy could be validated against the other models.

## 5. Results and Discussion

This section provides insights into the validation of the different energy models, their performance results, and the implications of these findings for practical applications in the field of battery electric bus planning.

### 5.1. Model Performance and Validation of the Framework

In this section, the accuracy of the propulsion and auxiliary models is analyzed and compared to generate the results of the presented framework utilized. For every single line from the tracking data, only input parameters for the EG, plus the actual energy demand,

were extracted. The input variables are only the day, time, and bus line. Based on this input, the EG provides input for the different energy models. A best-case model was included for both cases, propulsion and auxiliary, indicating the best prediction performance, having complete knowledge of the environment variables. For the propulsion case, the reference model was the physical energy model based on the tracking data. For the auxiliary case, the MLP on the test dataset is the reference model.

Our analysis revealed varying degrees of accuracy across different models. The Constant Model, being the simplest, had the highest Mean Absolute Error (MAE) and MAPE. Incorporating altitude data into the Constant Model significantly improved its accuracy, reducing the MAE and error percentage. The Daytime Model with Altitude offered further improvements, albeit marginally. It becomes clear that there is a big gap between the best model utilizing the EG and the reference model, which is to be expected since the EG is far from perfect. Please refer to Table 3 for the detailed values. Future studies could investigate other approaches using the physical model with a traffic simulation.

**Table 3.** Model performance metrics for the different propulsion models.

| Model | MAE (All Data) | MAPE |
|---|---|---|
| Constant Model | 0.524 kWh/km | 46.7% |
| Constant Model + Altitude | 0.148 kWh/km | 13.2% |
| Daytime Model + Altitude | 0.145 kWh/km | 12.9% |
| Physical Energy Model with tracking data * | 0.062 kWh/km | 5.5% |

Note: the * denotes the baseline model with optimal input data from historic trips.

In terms of the Auxiliary Models, similar trends were observed. The Constant and Monthly Constant Model showed a significant MAPE, while the Temperature and MLP Model offered better accuracy. Since we know that the labeling of the data points is not perfect in regards to diesel heating usage, we decided to validate the models separately on the complete dataset and only the data above 10 °C, since we know that there are no side effects from the diesel heating. Find the detailed results for this experiment in Table 4 In a follow-up study, this limitation should be eliminated. With correctly labeled data, it would be possible to generate better models for buses with and without diesel heating.

This division shows the models' difficulties in coping with the uncertainty in the lower temperature ranges, whether additional diesel heating was used or not. As with the propulsion model, we can also see that perfect input data significantly improves the overall results, which should motivate more research into generating better and more accurate input data for the models.

**Table 4.** Model performance metrics for the different auxiliary models on two different datasets: the whole dataset and for all data where the temperature is above 10 °C.

| Model | MAE (All Data) | MAPE | MAE (Below 10 °C) | MAPE |
|---|---|---|---|---|
| Constant Model | 0.243 kWh/km | 42.6% | 0.225 kWh/km | 39.5% |
| Monthly Constant | 0.223 kWh/km | 39.1% | 0.157 kWh/km | 27.5% |
| Temperature Model | 0.222 kWh/km | 38.9% | 0.146 kWh/km | 25.6% |
| MLP Model | 0.223 kWh/km | 39.1% | 0.145 kWh/km | 25.4% |
| MLP with tracking data * | 0.091 kWh/km | 16.0% | 0.0725 kWh/km | 12.7% |

Note: the * denotes the baseline model with optimal input data from historic trips.

Compared with the existing methods in the literature, our models with perfect input data are on par with the existing models, having a combined MAPE of 7.9%. But, when considering the case of predicting values for future bus trips, they significantly outperform models with constant values. Traditional models often rely on average consumption values,

which can lead to significant inaccuracies in diverse operational conditions. More complex models often ignore the fact that it is hard to predict future operational conditions.

### 5.2. Implications for Electric Bus Fleet Management and Optimization

The results of this study have significant implications for the management and optimization of electric bus fleets. Accurate energy consumption forecasting enables more efficient route planning, charging schedule optimization, and overall operational efficiency. It also aids in reducing operational costs by minimizing the need for large safety margins in energy planning, contributing to a more sustainable and economically viable transition to electric public transport systems.

In order to make reliable energy predictions and ensure the feasibility of the bus operations, we used a Monte Carlo simulation. We performed fit normal distributions to the energy demands in summer and winter. Since the vehicles serve 20 lines during the daily operations, the important information for the operator is whether the bus stays within its predicted energy demand for the total daily operation. This can be ensured with the fitted distributions and a Monte Carlo simulation. When using the 80th percentile of the energy demand for the winter case, the probability of exceeding the predicted values in sum over the whole day (20 trips) is below 99.9%, meaning less than once in two years (see Figure 15.

This approach is the basis for the following scenarios. Two scenarios were analyzed:

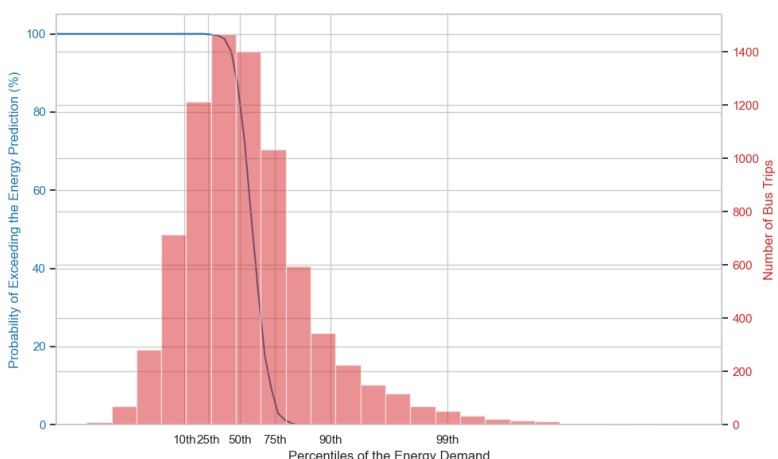

**Figure 15.** Histogram of the energy demand for the bus trips. In blue, it is the probability of the vehicle exceeding the predicted energy demand during the course of 20 trips when using different percentiles of the energy demand.

### 5.2.1. Scenario 1: Depot Charging

With a daily mileage of 200 km and a fleet of 100 buses, the traditional consumption assumption of 2.5 kWh/km, as suggested by the manufacturer, results in a total energy requirement of 50 MWh, equating to battery costs of 6.8 million euros (assuming battery prices stay at 136 EUR/kWh [66]). However, by adjusting the consumption estimates to 1.8 kWh/km in winter and 1.56 kWh/km in summer, the actual energy demands are statistically exceeded only once every two years. The total energy requirement drops to 36 MWh, reducing the battery cost to 4.9 million EUR. This approach ensures more than 99.9% confidence for the vehicles to serve their trips in both seasons.

### 5.2.2. Scenario 2: Lunch-Time Charging

For a split daily operation with charging once during the afternoon (100 km before and after), the change in consumption rates significantly impacts the charging demand and battery capacity requirements. The charging demand per vehicle drops from 250 kW

to 180 kW. Consequently, the total required battery capacity for the fleet decreases from 25 MWh to 18 MWh, reducing battery costs from 3.4 million EUR to 2.4 million EUR.

These findings have practical implications for the deployment and management of BEBs. Improved energy forecasting translates into more reliable and efficient public transportation services, aiding transit authorities in achieving sustainability goals and enhancing passenger experiences while highlighting the potential for substantial cost savings via a more nuanced understanding of energy consumption patterns in electric bus fleets.

### 6. Conclusions

This research started out with an in-depth data analysis of 16 battery electric buses operated for 13 months in Göttingen, Germany. The analysis revealed interesting findings about the different influencing factors for the energy demand of the vehicles. Based on those findings, data-driven models for the auxiliary and propulsion energy demand were developed and validated against the measured data from the buses. Together with the environment generator, which provides the models with the necessary input for the energy prediction, like passenger volume, weather conditions, and traffic, this presents a novel energy forecasting framework.

While there is still room to improve the different parts of this framework, it forms a solid basis for analyzing the strengths and weaknesses of different models and helps in structuring the forecasting. There are big differences in the accuracy of the different models; more complexity does not always mean better planning of bus operations in the long term. This is why we think this energy prediction framework makes an important contribution to the field of BEB planning by making the models more comparable.

Our study ventured to answer the previously posed research questions in energy forecasting for battery electric buses (BEBs). The findings are summarized as follows:

- **Can data-driven approaches improve energy forecasting for battery electric buses over constant value assumption for practical applications?**
  Yes, our research clearly demonstrates that data-driven approaches markedly improve energy forecasting for BEBs. The incorporation of real-time data, such as altitude, temperature, and passenger load, into our models significantly enhances the accuracy of predictions compared to traditional constant value assumptions. This improvement is crucial for operational efficiency and strategic planning in practical applications.
- **How big is the error margin for data-driven models versus constant values?**
  The error margin for data-driven models is substantially lower than that for models based on constant values. In our study, the MAPE for data-driven models was significantly lower. For the propulsion models, the MAPE was reduced from 46.7% for the constant model to 12.9% for the daytime model. For the auxiliary models, the MAPE was reduced from 39.5% for the constant model to 25.4% for the MLP model. This reduction in the error margin underlines the efficacy of data-driven approaches in capturing the complex dynamics of BEB energy consumption.
- **How can bus operators benefit from more precise energy forecasting?**
  Bus operators stand to gain considerably from more precise energy forecasting. Firstly, it allows for more efficient route and charging schedule planning. Secondly, it can lead to cost savings by reducing the need for large batteries. Finally, accurate forecasting supports the broader objective of sustainable urban transit by facilitating the effective integration and operation of BEBs in public transport networks.

The study confirms the benefits of integrating specific operational data into energy consumption models for BEBs. The advanced predictive capabilities of our models highlight the potential of data-driven approaches in this field.

While our study offers important insights into forecasting models for BEBs, certain constraints naturally limit its scope. One notable limitation is the models' performance across different geographical and climatic conditions, which presents an interesting ground for future research. This exploration could focus on how regional variations impact the efficacy

of each model, as discussed in Section 5.1. Future studies could aim to systematically assess and compare the performance of these models in diverse environmental settings, offering more nuanced and context-specific insights. Furthermore, the rapidly evolving landscape of data sources and technological advancements offers another promising direction for future research. Upcoming studies could explore the integration of emerging technologies and novel data sources into existing forecasting models. This could include harnessing AI and machine learning advancements, IoT-enabled data collection, and real-time analytics. Investigating these integrations could potentially lead to the development of more sophisticated and accurate forecasting models, further enhancing the reliability of BEBs' operational planning.

**Author Contributions:** Conceptualization, S.W., K.B., U.G. and A.R.; methodology, S.W.; software, S.W.; validation, S.W. and A.R.; formal analysis, all authors; investigation, S.W.; resources, S.W.; data curation, S.W.; writing—original draft preparation, S.W.; writing—review and editing, all authors; visualization, S.W.; supervision, A.R.; project administration, A.R.; funding acquisition, A.R. All authors have read and agreed to the published version of the manuscript.

**Funding:** This research received no external funding.

**Data Availability Statement:** The aggregated line data of the BEBs used for this study can be accessed online: https://osf.io/y7gr2/ (accessed on 10 December 2023).

**Conflicts of Interest:** The authors declare no conflicts of interest.

## Abbreviations

The following abbreviations are used in this manuscript:

| | |
|---|---|
| ANN | Artificial Neural Network |
| API | Application Programming Interface |
| BEB | Battery Electric Bus |
| EG | Environment Generator |
| GPS | Global Positioning System |
| HVAC | Heating Ventilation Air Conditioning |
| IQR | Inter Quartile Range |
| kW | Kilowatt |
| kWh | Kilowatt Hours |
| MAE | Mean Absolute Error |
| MAPE | Mean Absolute Percentage Error |
| MLP | Multilayer Perceptron |
| NN | Neural Network |
| OEM | Original Equipment Manufacturer |
| pkm | Passenger Kilometers |
| POI | Point of Interest |
| RMSE | Root Mean Squared Error |
| SOC | State of Charge |

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
