# Peer review of "Towards Efficient Battery Electric Bus Operations: A Novel Energy Forecasting Framework"

_wevj, doi:10.3390/wevj15010027_

Round 1

Reviewer 1 Report

Comments and Suggestions for Authors

This paper introduces an innovative forecasting methodology that combines a propulsion and auxiliary energy model with a novel concept, the environment generator. This approach addresses the primary challenge in electric bus energy forecasting: estimating future environmental conditions, such as weather, passenger load, and traffic patterns, which significantly impact energy demand. Graph quality is good. The overall paper is well organized and well written. This reviewer advice the acceptance of this paper. 

Author Response

Thank you for taking the time to review my manuscript.

Thank you for recomending the acceptance of the paper.

Reviewer 2 Report

Comments and Suggestions for Authors

Here's my evaluation:

Originality:

The paper introduces a novel energy forecasting methodology for battery electric buses (BEBs). The methodology integrates a propulsion and auxiliary energy model with an environment generator. It is unique in its comprehensive treatment of variables such as weather, passenger load, and traffic patterns, which are critical for accurate energy prediction in electric bus operations. The paper's originality is enhanced through the use of real-world operational data from a case study in Göttingen, Germany.

Language:

The language used in the paper is clear, professional, and suitable for an academic journal. Technical terms are appropriately defined and used, ensuring comprehensibility for readers familiar with the field. The structure of the paper, including the introduction, methodology, results, and conclusion, is logically organized and well-articulated.

Specific Comments for Revision:

Clarification of Methodological Details: The paper could benefit from more detailed explanations of the statistical methods and algorithms used in the study. This includes a deeper dive into the environment generator's functioning and its integration with other models.

Comparative Analysis: While the paper does a good job of validating the proposed model, a more explicit comparison with existing models would strengthen the argument for its novelty and effectiveness.

Impact Assessment: The discussion on the practical implications of the findings, particularly in the context of urban transport policy and electric bus fleet management, could be expanded to add value.

Graphical Representations: Some of the figures and graphs could be more clearly labeled and interpreted within the text. Ensuring that these visual aids effectively complement the written content is crucial.

Future Research Directions: The conclusion can be expanded to suggest specific directions for future research, building on the findings and limitations of the current study.

Author Response

I appreciate the time and effort you have dedicated to providing valuable feedback on my manuscript. I am grateful for your insightful comments on my paper. I have been able to incorporate changes to reflect most of the suggestions provided. I have highlighted the changes within the manuscript.

Here is a point-by-point response to your comments and concerns.

  1. Comment: Clarification of Methodological Details: The paper could benefit from more detailed explanations of the statistical methods and algorithms used in the study. This includes a deeper dive into the environment generator's functioning and its integration with other models.

Thank you for your comment. While I agree that some of the described methodologies are rather technical, in order to not overload the manuscript, we refrained from describing standard statistical tools like analyzing mean, min, and max values as well as the different percentiles of the data. Nonetheless, I included remarks in the introductory text for the subsection “Influencing Factors” and “Environment Generator” to make it clearer. Thank you for this remark. If you have concrete passages needing clarification, I would happily address them.

  1. Comment: Comparative Analysis: While the paper does a good job of validating the proposed model, a more explicit comparison with existing models would strengthen the argument for its novelty and effectiveness.

Thank you for this comment. We provided the results of different models in the literature section and presented our results, which should help us understand the quality of the presented models. Since it is impossible to reproduce the results of other papers, especially for the deep learning-based models, we decided on the described approach.

  1. Comment: Impact Assessment: The discussion on the practical implications of the findings, particularly in the context of urban transport policy and electric bus fleet management, could be expanded to add value.

We provided two realistic examples of different bus operations, including assessing the implications when using our models compared to the often-used constant values (represented by our constant models). While a more in-depth discussion of implications could be interesting, it is outside of the scope of this paper.

  1. Comment: Graphical Representations: Some of the figures and graphs could be more clearly labeled and interpreted within the text. Ensuring that these visual aids effectively complement the written content is crucial.

While I appreciate your effort in reviewing the figures, it would be helpful to know which figures need revision. Since we already tried to make them as expressive as possible.

  1. Comment: Future Research Directions: The conclusion can be expanded to suggest specific directions for future research, building on the findings and limitations of the current study.

I Agree with this comment. The last paragraph was revised and expanded to illustrate different research directions.

I hope I addressed all your comments in enough detail. And I am happy to provide more information if required. I want to thank you again for your time and effort!

Reviewer 3 Report

Comments and Suggestions for Authors

The strength of this work is that it compares and find new effective solutions for energy forecasts for application battery electric buses. It is nowadays the main topic of energy issues besides energy storage.

1.       In row 69, appear the first time the term: Environment generator (EG) concept. The review doesn´t find in text the definition of EG. What it exactly means? It is an important part of the current Research.

2.       In the row 126, is the term 12m? What it means?

3.       In the row 128-129 claimed that fossil fuels get 128 reduced by 32-56% by utilizing BEBs instead of diesel buses. In what conditions? Why only 32-56%, not 100%?

4.       The signature of Fig. 3 might be shorter. It's enough, only one first sentence. Others might be part of the main text.

5.       In row 502 in word, DEspite is not necessary to place e as a capital letter.

6.       In the signature of Fig. 13, it is said that the red line is the mean value for cold temperatures without the additional diesel generator. As it reality seems, the diesel generator is mandatory in lower temperatures.

7.       In row 601, the decision is made by Ockham Razor. What is it? No explanation.

8.       Is it acceptable, from the style of the article, that table headings are down the tables?

9.       In row 661, the authors said that the effect on auxiliary power is comparatively minor from passengers' volume. How much is it? Why not take into account the volume of passengers in cold periods? Heating inside the bus influences the volume of passengers significantly. The heat power of the average man is about 100 W.

Author Response

I appreciate the time and effort that you have dedicated to providing your valuable feedback on my manuscript. I am grateful for your insightful comments on my paper. I have been able to incorporate changes to reflect most of the suggestions provided. I have highlighted the changes within the manuscript.

Here is a point-by-point response to your  comments and concerns.

  1. In row 69, appear the first time the term: Environment generator (EG) concept. The review doesn´t find in text the definition of EG. What it exactly means? It is an important part of the current Research.

Thank you for pointing this out. I agree with this comment, although the concept is explained later on. I added a very short description at the mentioned row, to make it easier for the reader.

  1. In the row 126, is the term 12m? What it means?

12m bus stands for 12-meters bus. Which is one of the two most common length of buses in urban public transport. Thank you for pointing this out. For clarification I changed it to 12-meter.

  1. In the row 128-129 claimed that fossil fuels get 128 reduced by 32-56% by utilizing BEBs instead of diesel buses. In what conditions? Why only 32-56%, not 100%?

The cited study discusses the life-cycle fossil fuel consumption of a battery electric bus versus a diesel bus. Therefore, no concrete conditions are mentioned. I clarified this at the mentioned section of the manuscript. Thank you for this comment.

  1. The signature of Fig. 3 might be shorter. It's enough, only one first sentence. Others might be part of the main text.

Agree. I have shortened the text to make it more concise.

  1. In row 502 in word, DEspite is not necessary to place e as a capital letter.

Thank you for pointing out this typing error.

  1. In the signature of Fig. 13, it is said that the red line is the mean value for cold temperatures without the additional diesel generator. As it reality seems, the diesel generator is mandatory in lower temperatures.

Thank you for this comment. I don’t think that we can conclude from this analysis that additional diesel heating is mandatory to enable electric bus operations. I would consider it a safety measure from the manufacturer and operator. However, I am confident that in the future more and more buses will be operated purely electric and it is important to reflect this in forecasting models.

  1. In row 601, the decision is made by Ockham Razor. What is it? No explanation.

Thank you for pointing this out. I added a short explanation for  Ockham’s Razor or also revered to as Occam’s Razor and added a fitting citation.  

  1. Is it acceptable, from the style of the article, that table headings are down the tables?

I would assume so, since I did not change the template.

  1. In row 661, the authors said that the effect on auxiliary power is comparatively minor from passengers' volume. How much is it? Why not take into account the volume of passengers in cold periods? Heating inside the bus influences the volume of passengers significantly. The heat power of the average man is about 100 W.

Thank you for this comment. The increase is mentioned in the Data Analysis Section. And shown in Figure 9. The average consumption increases from 0.36 kWh/km to 0.39 kWh/km. By evaluating the models in conjunction with the environment generator, we found that the uncertainty in predicting the passenger volume did nullify the gained precision of incorporating the passenger volume in the auxiliary energy model. Therefore, in future studies, after improving the environment generator this could be an interesting improvement. We assumed that the increase in passengers correlates with the increase in stops and therefore door openings, which could reduce the heating effect of the passengers.

I hope I addressed all your comments in enough detail. And I am happy to provide more information if required. And want to thank you again for your time and effort!